# Fatty Acids and Lipid Paradox-Neuroprotective Biomarkers in Ischemic Stroke

**DOI:** 10.3390/ijms231810810

**Published:** 2022-09-16

**Authors:** Sebastian Andone, Lénárd Farczádi, Silvia Imre, Rodica Bălașa

**Affiliations:** 1Ist Neurology Clinic, Emergency Clinical County Hospital Targu Mures, 540136 Targu Mures, Romania; 2Department of Neurology, ‘George Emil Palade’ University of Medicine, Pharmacy, Science, and Technology of Targu Mures, 540136 Târgu Mures, Romania; 3Doctoral School, ‘George Emil Palade’ University of Medicine, Pharmacy, Science, and Technology of Targu Mures, 540142 Targu Mures, Romania; 4Center for Advanced Medical and Pharmaceutical Research, ‘George Emil Palade’ University of Medicine, Pharmacy, Science, and Technology of Targu Mures, 540136 Targu Mures, Romania; 5Department of Analytical Chemistry and Drug Analysis, Faculty of Pharmacy, ‘George Emil Palade’ University of Medicine, Pharmacy, Science, and Technology of Targu Mures, 540136 Targu Mures, Romania

**Keywords:** ischemic stroke, fatty acids, lipid profile, lipid paradox, neuroprotection

## Abstract

Stroke is the primary cause of death and disability worldwide, with ischemic stroke up to 80% of the total cases. Lipid profile was established as a major risk factor for stroke, but recent studies show a paradoxical relationship between serum values and the outcome of stroke patients. Our study aims to analyze the impact of the classic extended lipid profile, including fatty acids as potential neuroprotective biomarkers for the outcome of ischemic stroke patients. We included 298 patients and collected clinical, paraclinical, and outcome parameters. We used a method consisting of high-performance liquid chromatography coupled with mass spectrometry to quantify serum fatty acids. We observed a negative correlation between National Institutes of Health Stroke Scale (NIHSS) at admission and total cholesterol (*p* = 0.040; r = −0.120), respectively triglycerides (*p* = 0.041; r = −0.122). The eicosapentaenoic to arachidonic acid ratio has a negative correlation, while the docosahexaenoic to eicosapentaenoic acid ratio positively correlates with all the prognostic parameters, showing a potential neuroprotective role for eicosapentaenoic acid in preventing severe ischemic stroke. The impact of the lipid profile paradox and the dependency relationship with the fatty acids represent a significant predictive factor for the functional and disability prognostic of ischemic stroke patients.

## 1. Introduction

Stroke represents a clinically defined syndrome by a focal neurological deficit due to a vascular injury of the central nervous system. It represents one of the primary causes of death and disability worldwide, the ischemic stroke summing up to 80% of the total number of stroke cases. Stroke is not a unique pathology, as a diverse range of risk factors and physiopathological mechanisms can play a role in its incidence [1].

Dyslipidaemia is one of the general risk factors associated with cardiovascular and cerebrovascular diseases, being one of the critical elements in the development of atherosclerotic plaques.

The Framingham study has adequately established the role of the lipidic profile as a cardiovascular risk factor, and it is used in most clinical practice guidelines, including total cholesterol, triglycerides, and the high-density lipoprotein (HDL) cholesterol and low-density lipoprotein (LDL) cholesterol. From these, significant importance was attributed to the LDL fraction, considered the primary atherogenic lipoprotein [2,3].

Although a significant risk factor, the lipid profile was brought again into the spotlight because of recent studies showing a phenomenon called “reverse epidemiology”. This phenomenon observed in ischemic stroke patients showed a paradoxical relationship between the lipidic profile and functional outcome of patients [4].

Several hypotheses were stated regarding the physiopathological mechanism responsible for this paradox. Among these is the nutritional hypothesis, which states that patients with a higher nutritional reserve and, therefore, a higher serum level of cholesterol and triglycerides will have a favorable outcome compared to patients with a modest nutritional status [5].

Other possible mechanisms implied are the role in the maintenance of vascular integrity, the increased rupture resistance of the vessels, neuroprotective effects in the central nervous system, and neuroplasticity’s role in developing new synapses [6].

The fatty acids can be classified into three categories depending on their structural formula: saturated fatty acids, monounsaturated fatty acids (MUFA), and polyunsaturated fatty acids (PUFA). From these, Omega-3 PUFA includes alfa-linoleic acid (ALA), eicosapentaenoic acid (EPA), and docosahexaenoic acid (DHA), while Omega-6 PUFA includes linoleic acid (LA) and arachidonic acid (AA) [7]. ALA and LA are essential because they cannot be synthesized by the human organism and play a role in AA, DHA, and EPA metabolize. AA is a precursor for certain prostaglandins, thromboxane, and leukotrienes and has a preponderant anti-inflammatory role. EPA is also a precursor for prostaglandin-3, which inhibits platelet aggregation and has an anti-inflammatory role, as well as a role in the reduction of the parietal thickness of the arteries [8,9]. DHA and AA have an essential role in cerebral development and neurogenesis, as well as in signaling pathways, genic expression, and the structure and maintenance of the cellular membranes. They have a relevant role in neuroprotection against neurodegenerative pathologies such as Parkinson’s and Alzheimer which are associated with mitochondrial dysfunction, neuroinflammation, and oxidative stress [10]. DHA, EPA, and AA are poorly synthesized in the body, and several clinical trials proved that a rich diet in these fatty acids reduces the risk of cardiovascular disease [11].

During the ischemic stroke, the structure of the cellular membranes shifts and favors the destruction of the neuronal membrane phospholipids releasing at a cellular level arachidonic acid. Taking into account previous considerations suggested by certain authors, the aim of the study is the investigation of the serum ratio of Omega-3 and Omega-6 PUFA, such as EPA/AA and DHA/AA, quantified by high-performance liquid chromatography coupled with mass spectrometry (LC-MS/MS), as markers to determine the relationship between the lipid profile and the fatty acids profile related to stroke severity outcome [12].

## 2. Results

### 2.1. Study Population Analysis

There were 153 (51.3%) male patients and 145 (48.7%) female patients. The median age of the group is 69.93 ± 13.45. There is a significant statistical difference between the means of the female and male patients, the male patients’ age being 67.27 ± 14.14 while the female patients have an age of 72.74 ± 12.11 (*p* < 0.001).

The patients were admitted for 9.55 ± 6.72 days. The male patients required 8.40 ± 3.98 days of hospitalization while the female patients required 10.77 ± 8.57 days of hospitalization, a statistically significant difference (*p* = 0.002).

Of the total number of patients, 274 (91.9%) had a cerebral infarction, while 24 (8.1%) presented a transient ischemic attack (TIA). At admission, 61 patients (20.5%) received revascularization treatment (thrombolysis, thrombectomy, or both).

During hospitalization, 13 (4.4%) patients presented hemorrhagic transformation. Grouped by the territory of vascularization affected, 242 (81.2%) of the patients had a stroke in the carotid territory, while 56 (18.8%) had a vertebrobasilar stroke. As far as it concerns the subtype of stroke based on TOAST classification, 49 (16.4%) of the patients had large vessel disease (LVD), 28 (9.4%) had small vessel disease (SVD), 89 (29.9%) had cardioembolic etiology (CE), 12 (4.0%) had other determined etiologies. In comparison, 120 (40.3%) had unknown etiology (at least two primary etiologies being excluded).

All the data mentioned above are summarized in Table 1. Paraclinical findings of the patient group can be found summarized in Table 2.

### 2.2. Clinical Data Regarding the Functional Deficit, Disability, and Mortality

At admission, patients had a previous modified Rankin scale score of 0.24 ± 0.68 points and an admission NIHSS of 6.75 ± 5.76 points.

At discharge, the modified Rankin scale score was 2.50 ± 1.85 points and an NIHSS of 4.28 ± 4.55 points. Grouped by the stroke severity at admission, 112 (40.9%) patients were in the low-severity subgroup, while 162 (59.1%) were in the moderate-severe subgroup.

At discharge, 169 (66.3%) of the patients were in the low-severity subgroup, while 86 (33.7%) were in the moderate–severe subgroup.

During hospitalization, 19 (6.4%) patients died.

The patients’ distribution and pathways between admission and discharge regarding NIHSS can be observed in Figure 1.

All these data can be found summarized in Table 3.

### 2.3. Sex Subgroup Analysis

Except for the already mentioned results regarding age and hospitalization days, there were several statistically significant differences between the male and the female patients for other parameters, which will be listed below and summarized in Table 4 and Table 5.

DHA/EPA ratio was 2.84 ± 0.88 for the male patients and 3.15 ± 1.00 for the female patients (*p* = 0.005). As for the lipidic profile, the total cholesterol was 180.32 ± 42.82 respectively, the HDL-c was 45.61 ± 11.86 in the male subgroup, while in the female subgroup, the total cholesterol was 194.87 ± 56.35 and the HDL-c was 53.20 ± 17.60. The differences between the two subgroups were statistically significant for total cholesterol (*p* = 0.013) and HDL-c (*p* = 0.027).

Another paraclinical parameter with a statistically significant difference was hemoglobin. The males had a hemoglobin of 14.42 ± 1.49, while the females had a hemoglobin of 13.10 ± 1.72 (*p* < 0.001).

Regarding the prognostic and disability factors, we observed a difference in the NIHSS at discharge. The males had 3.74 ± 0.99 points while the females had 4.85 ± 5.04 (*p* = 0.042). There were no significant differences between the male and the female subgroups regarding the modified Rankin scale at admission and discharge and the NIHSS at admission. (Table 5). However, there was a statistically significant association between the subgroups of patients grouped by sex and severity of stroke at admission. (*p* = 0.050) (Table 6).

### 2.4. Ischemic Stroke Type Subgroup Analysis

The patients with a cerebral infarct had a mean age of 70.53 ± 12.63 years and required 9.92 ± 6.86 days of hospitalization, while the patients with TIA had a mean age of 63.08 ± 19.77 years and required 5.42 ± 1.99 days of hospitalization. There was a statistically significant difference between the two subgroups in age (*p* = 0.009) and the required hospitalization days (*p* = 0.002).

Another observed difference was the DHA/EPA ratio which was 3.04 ± 0.95 in the infarct subgroup and 2.39 ± 0.71 in the TIA subgroup (*p* = 0.001) (Table 4).

### 2.5. Vascularization Territory Subgroup Analysis

The patients with a carotid stroke had a mean age of 71.30 ± 13.28 and required 10.00 ± 7.27 days of hospitalization, while the patients with vertebrobasilar stroke had a mean age of 64.04 ± 12.69 and required 7.63 ± 2.77 days of hospitalization. There were statistically significant differences both for age (*p* < 0.001) as well as for the required hospitalization days (*p* = 0.017) between the two subgroups (Table 4).

From the paraclinical data, only hemoglobin showed a statistically significant difference, the carotid stroke patients having hemoglobin of 13.68 ± 1.76 while the vertebrobasilar stroke patients had a hemoglobin of 14.2 ± 1.52 (*p* = 0.042).

Regarding the functional and prognostic factors, there were extremely significant differences between the two subgroups for mRS at discharge (*p* < 0.001), NIHSS at admission (*p* < 0.001) as well as NIHSS at discharge (*p* < 0.001) (Table 5).

Additionally, significantly statistical associations were found between the subgroups of patients grouped by the affected territory and the severity of stroke at admission (*p* < 0.001) and between the subgroups grouped by the affected territory and the severity of stroke at discharge (*p* = 0.001) (Table 6).

### 2.6. Hemorrhagic Transformation Subgroup Analysis

There was an extremely statistically significant difference between patients who presented a hemorrhagic transformation requiring 20.8 ± 11.96 days of hospitalization and those that did not have such a complication requiring 9.07 ± 5.99 days of hospitalization (*p* < 0.001).

From the paraclinical data, only the leucocyte count differed between the two subgroups (*p* = 0.001) (Table 4).

There were statistically significant differences regarding mRS at discharge (*p* = 0.001), NIHSS at admission (*p* = 0.002) as well as NIHSS at discharge (*p* = 0.001) (Table 5).

The association between the subgroups of patients grouped by hemorrhagic transformation and the severity of stroke at admission (*p* = 0.017) and the association between the subgroups of patients grouped by hemorrhagic transformation and the severity of stroke at discharge (*p* = 0.005) were statistically significant.

### 2.7. TOAST Subgroup Analysis

In the case of the patients with large vessel disease (LVD), there was a statistically significant difference in glycemia at admission, the patients with LVD had a value of 145.03 ± 90.18 mg/dL, while the rest of the patients had a value of 126.07 ± 42.63 (*p* = 0.024) (Table 4). Regarding this etiology, there were no statistically significant differences for other parameters, including prognostic parameters (Table 5). Additionally, there were no associations between the subgroups of patients grouped by LVD and stroke severity at admission or discharge (Table 6).

For patients with small vessel disease (SVD), the only statistically significant differences were for mRS at discharge (*p* = 0.014) and NIHSS at admission (*p* = 0.006) but not for NIHSS at discharge (*p* = 0.055).

As for the cardioembolic etiology, we observed an extremely significant difference; the patients with cardioembolic stroke had a mean age of 75.99 ± 11.96 years, while the rest had a mean age of 67.35 ± 13.25 years (*p* < 0.001). Also for this etiology, we observed statistically significant differences for the following paraclinical parameters: total cholesterol (*p* = 0.005), triglycerides (*p* = 0.002), hemoglobin (*p* < 0.001), and INR (*p* < 0.001).

Regarding the prognostic parameters, we found statistically significant differences for the cardioembolic stroke in mRS at discharge (*p* = 0.008), NIHSS at admission (*p* < 0.001), and NIHSS at discharge (*p* = 0.030) (Table 5).

Statistically significant associations were observed between the subgroups of patients grouped by the severity of stroke at admission with the subgroups of patients with SVD (*p* = 0.018) and with subgroups of patients with cardioembolic stroke (*p* = 0.008). No associations were found with the subgroups of patients based on the stroke severity at discharge (Table 6).

### 2.8. Stroke Severity Subgroup Analysis

For both the subgroups of patients grouped by the severity of stroke at admission as well the severity of stroke at discharge, we found three parameters that had statistically significant differences between each subgroup, as follows: age (*p* = 0.001 respectively *p* = 0.026), hospitalization days (*p* < 0.001 respectively *p* < 0.001) and DHA/EPA ratio (*p* = 0.043 respectively *p* = 0.012) (Table 4).

### 2.9. Hospitalization Mortality Subgroup Analysis

The deceased patients had a mean age of 80.11 ± 9.58 years, while the survivor subgroup had a mean age of 69.24 ± 13.41 (*p* = 0.001). There were statistically significant differences between the deceased and the survivor subgroups regarding leucocyte count (*p* = 0.015) and hemoglobin (*p* = 0.011) (Table 4).

In the case of the deceased patients, we had a previous mRS of 0.95 ± 1.31 points and an NIHSS at the admission of 13.37 ± 3.83 compared with the survivor subgroups where the mRS before admission was 0.19 ± 0.59 points, and the NIHSS at admission was 6.30 ± 5.60. Both parameters had extremely significant differences between the two subgroups (*p* < 0.001 respectively *p* < 0.001) (Table 5).

### 2.10. Bivariate Correlations between Clinical, Paraclinical, and Prognostic Parameters

All the correlations between clinical, paraclinical, and prognostic parameters are summarized in Table 7, Table 8, Table 9 and Table 10.

Nevertheless, some correlations are worth mentioning, like EPA/AA ratio was negatively correlated with mRS at discharge (*p* = 0.020), with NIHSS at admission (*p* = 0.023) and NIHSS at discharge (*p* = 0.007); DHA/EPA ratio positively correlated with mRS at discharge (*p* < 0.001), NIHSS at admission (*p* < 0.001) and NIHSS at discharge (*p* < 0.001); total cholesterol and triglycerides, both negatively correlated with NIHSS at admission (*p* = 0.040 respectively *p* = 0.041) (Table 7).

### 2.11. Stroke Severity at Discharge Prediction Model

Omnibus tests of model coefficients were significant. Chi-square = 130.326, *p* < 0.001.

The model’s goodness of fit was tested with the Hosmer–Lemeshow test (*p* = 0.089), proving a good fit for the model.

The model explains 55.5% of the variance of the stroke severity at discharge and correctly predicts 82.4% from the cases with a sensitivity of 76.7% and a specificity of 85.2%.

The age, sex, vascularization territory, mRS before admission, DHA, EPA, DHA/AA, EPA/AA, and AA/(DHA + EPA) variables proved no significance in the model. The patients who underwent revascularization therapy had a higher chance of being in the low-severity subgroup (OR = 0.383; *p* = 0.021). Additionally, patients grouped in the low severity subgroup at admission were highly likely to be present in the same subgroup at discharge (OR = 0.020; *p* < 0.001). Additionally, patients that develop hemorrhagic transformation have a higher chance of more than 6.954 times being in the moderate-severe subgroup at discharge (*p* = 0.046).

The DHA/EPA ratio increase is also associated with a higher chance of the patients being included in the moderate-severe subgroup at discharge (OR = 2.207, *p* = 0.029), while AA serum levels increase associated with a minor increase in change to be included in the moderate–severe subgroup at discharge (OR = 1.009; *p* = 0.018).

The model was summarized in Table 11, and the Forrest plot of the model can be observed in Figure 2.

## 3. Discussion

The study aimed to analyze the impact of the lipid profile and the serum fatty acids panel (including AA, DHA, EPA, and their ratios) over the functional, disability, and mortality outcomes in ischemic stroke patients.

The study population is somehow homogenous from the view of sex distribution, with the number of females and males almost equal. However, we observed that the male patients were younger than the females; this could be explained by the predisposition of the male sex for cardiovascular events.

The female patients required prolonged hospitalization and had increased NIHSS at discharge, compared to male patients, the two parameters being dependent on one another. Even if males are predisposed to stroke at a younger age, the females need a more extended period of recovery in the acute phase and remain with worse neurological deficits. Of course, the female patients’ group had an increased age mean, which could also explain the increased hospitalization period, given the direct relationship between required hospitalization days and the age of the patients regardless of gender. We also noticed that female patients had higher total cholesterol and HDL-c values than male patients.

Other authors already established the gender difference regarding the lipid profile in the literature, as females tend to be more predisposed to dyslipidemia than men [13].

Hemoglobin was lower in the female patients’ group. This could be explained by several factors including that the hemoglobin was inversely proportional with age and because the female group had an older age. The evidence that females had a higher NIHSS at discharge can also be due to the influence of the lower hemoglobin in this group, which had a reversed effect on the functional outcome.

Another observed difference between males and females was regarding DHA/EPA ratio, which was increased in the female group. Another exciting aspect of this ratio was that higher values were found in the moderate–severe stroke group at admission and discharge. Besides this, the group of patients with an infarct also had higher values than the TIA group. As established by the predictive model of stroke severity at discharge, an increased value of this ratio multiplies the chance for increased severity functional outcome.

Apart from this, we also found already established outcome predictors [14], such as revascularization and baseline stroke severity at admission. An increase in AA serum values slightly increases the chance of a more severe stroke at discharge.

In order to understand all these relationships better, there are a couple of explanations that are required.

DHA/EPA ratio positively correlates with mRS at discharge, NIHSS at admission, and NIHSS at discharge. An opposed situation is in the case of the EPA/AA ratio, which correlates negatively with the previously mentioned prognostic parameters. Even so, independent values of each analyzed fatty acid had no correlations with prognostic factors but had interdependent correlations, all three being directly proportional.

Given that there is a direct dependent relationship with the NIHSS and with functional outcome, to manifest a neuroprotective effect for patients at risk of developing an ischemic stroke, a viable option would be to try and decrease the DHA/EPA ratio. Two methods could achieve this: reduce the plasma’s DHA value or increase the EPA value. Considering that the EPA/AA ratio is also inversely proportional to the prognostic factors, the best solution to influence both ratios would be to increase the EPA value from the plasma.

AbuMweis et al. focused on a meta-analysis of the rapport between EPA and DHA in Omega-3 food supplements and observed that the higher the EPA value was compared to DHA, the lower the C-reactive protein value was. Additionally, blood pressure was lower when the EPA value was higher than DHA in the food supplements. This proves that we can influence the modifiable cardiovascular risk factors by changing the ratio between EPA and DHA [15].

Shojima et al. also established that a lower EPA/AA ratio at admission is associated with a poor long-term outcome and increased mortality in ischemic stroke patients. Besides this, a lower ratio was also associated with major cardiovascular events in this patient group [16].

Nelson et al. showed that EPA/AA ratio could be used as a marker for cardiovascular events. They also emphasized the importance of Omega-3 food supplements, proving that higher purity EPA supplements can improve this ratio and be associated with better clinical outcomes [17].

Ueno et al. identified a lower EPA/AA ratio in patients with multiple cerebral infarcts and white matter lesions. All these observed things from the literature are also relevant to our study, supporting, even more, our results [18].

Based on the territory of vascularization affected, we observed that patients with carotid stroke had higher NIHSS at admission and even higher NIHSS at discharge compared to patients with vertebrobasilar stroke. Even if a calculation bias exists for NIHSS in the case of vertebrobasilar stroke deficits, it is clear that the carotid stroke subgroup has a much worse outcome than the vertebrobasilar subgroup because mRS is also higher in the carotid group. In the case of this variable, there is no calculation bias related to the different clinical manifestations.

The patients with hemorrhagic transformation presented a two folded value regarding the necessary hospitalization days and had an increased level of leukocytes at admission compared to the patient that did not present this complication. The relationship between hemorrhagic transformation and leukocytosis has been approached in the literature. It appears that leukocytosis without the presence of an infectious cause represents a systemic marker of inflammation. The activation of leukocytes in the acute phase of the stroke produces an increased blood–brain barrier permeability. Antoniazzi et al. proved this on a cohort of over 200,000 patients with ischemic stroke, establishing that peripheral leukocytosis represents an independent prediction factor for hemorrhagic transformation and death during hospitalization [19]. This was also observed in our study population, where leukocytes were elevated in deceased patients compared to survivors.

Regarding TOAST classification, some results are worth mentioning and should be discussed. Patients with LVD have an increased value for glycemia at admission. Of course, we must admit that there is the possibility of a design bias; glycemia was collected at the arrival in the Emergency Department, in any period of the day, regardless of fasting status. Even so, we must mention that glucose-altered tolerance represents a risk factor for this group of patients for ischemic stroke.

Patients with SVD had a lower mRS at discharge and a lower NIHSS at admission than the rest. Meanwhile, cardioembolic stroke patients had an increased mRS at discharge and an increased NIHSS at admission and discharge than the rest. This is consistent with the data found in the literature, including the Framingham study [20].

Patients with cardioembolic stroke are older than the rest, which could be explained since cardiovascular disease incidence increases with age. The patients with an increased stroke severity at admission and discharge had older age and required a higher number of hospitalization days.

Regarding lipid profile, besides the already mentioned facts, we found a negative correlation between total cholesterol and DHA/EPA ratio, as well as a negative correlation between total cholesterol and NIHSS at admission. This strengthens the idea that the DHA/EPA ratio is directly proportional to the NIHSS score. There is also a directly proportional relationship between total cholesterol and LDL cholesterol, which does not apply to HDL cholesterol. There is, of course, a direct positive relationship between cholesterol and triglycerides. Triglycerides also have a positive correlation with LDL-c and a negative one with HDL-c. Similar to total cholesterol, we observed an inverse relationship between triglycerides and the NIHSS at admission. The HDL-c and LDL-c fractions do not maintain any correlation with prognostic variables. However, HDL-c has a positive correlation with DHA/AA ratio.

Cheng et al. showed that a low value of triglycerides and cholesterol in the acute phase of ischemic stroke is associated with increased mortality in statin naïve patients [21].

Jain et al. also showed that triglycerides are associated with a worse prognostic in ischemic stroke [22].

Koton et al. showed that low total cholesterol at admission was associated with increased stroke severity regardless of the statin treatment before the event [23].

Beltowski emphasized the neuroprotective effect created by the higher cholesterol value as it was associated with reduced long-term mortality in ischemic stroke [4]. This was also confirmed by Patel et al., who showed a paradoxical relationship between improved prognostic and reduced post-stroke complications in stroke patients with increased lipid profile levels [24].

Zhou et al. observed an association between a lower LDL-c and a higher risk of hemorrhagic stroke [25]. This was also found by Bharosay et al., but for lower cholesterol and triglycerides associated with increased hemorrhagic stroke and neurologic worsening of the patients with ischemic stroke [26].

### Limitations

This study has several limitations and shortcomings.

Firstly, this is a single-center study, so we cannot generalize our findings, as the study size should be increased to strengthen the conclusions and confirm the results. The functional outcome of the patients can be affected by multiple factors which could not be analyzed in this study. We also have to mention that it would have been helpful to have a follow-up complementary study to observe the long-term prognostic factors.

One of the study’s limitations derives from the sample profile selection, as we included all patients regardless of their previous treatment (statins, antiplatelet, anticoagulation), regardless of the treatment received (thrombolysis, thrombectomy, or none of this).

We must mention that the NIHSS score contains a calculation bias that may partially explain the vertebrobasilar subgroup’s lower value. This is because the clinical manifestation in the vertebrobasilar stroke, for example, isolated cranial nerves palsies, nystagmus, vestibular syndrome, and specific clinical features of the cerebellar syndrome are not quantifiable by this score and therefore can create a false image of the severity of the stroke in this particular group of patients.

Another limitation of the study is related to the design of the two subgroups of stroke severity at admission and discharge. We included all patients with NIHSS less or equal to 4 in the lower group. However, as mentioned before, there was a possibility of having small NIHSS with severe neurological deficits in the case of vertebrobasilar strokes. The same could happen for cases where patients had only specific neurologic deficits like (hemianopsia and severe aphasia) with low NIHSS scores but severe deficits that should have been included in the severe subgroup.

## 4. Materials and Methods

### 4.1. Study Population Definition

We undertook a prospective observation study covering ischemic stroke patients admitted to 1st Neurology Clinic, Emergency Clinical County Hospital, Târgu Mureș.

**Inclusion criteria**: We included all admitted adult patients diagnosed with acute ischemic stroke (less than 72 h from the onset of the stroke), regardless of their previous treatment (antiplatelet, statins) and regardless of if they received revascularization therapy (thrombolysis or thrombectomy)

**Exclusion criteria**: We excluded all the patients with hemorrhagic stroke or stroke-mimic pathologies that were proven after the admission of the patients.

In the study period, over six months (January 2022–June 2022), we gathered a number of 321 consecutive patients with a stroke diagnosis. Six patients have been excluded from the study because they presented other stroke mimic pathologies. Of the rest, 17 presented hemorrhagic stroke and were excluded from the target group, leaving a final number of 298 patients with ischemic stroke.

The on-call neurologist established the diagnosis of ischemic stroke in the Emergency Department based on the medical history, clinical, paraclinical, and imaging criteria (CT scan), with the evolution follow-up of these parameters by the staff in the clinic.

The study protocol was in accord with the Helsinki Declaration and was approved by the ethics committee of the Emergency Clinical County Hospital (no. 28763/13.11.2018).

All patients (or legal guardians/family members) had signed a basic written consent form for allowing the data collection and the blood sample collection.

### 4.2. Data Collection

We collected clinical, paraclinical, and personal data from the files of the patients admitted.

The data was recorded in a specialized research database being exported for processing at the end of the recruitment phase. Data was reviewed before the final statistical analysis to increase the collected data’s accuracy.

We used the following scales: National Institutes of Health Stroke Scale (NIHSS) is a validated, systematic, quantitative scale used to measure neurological deficit in stroke patients [27]; the modified Rankin scale (mRS) is a scale used to measure the degree of disability in stroke patients in an interval from 0 to 6 [28].

We used the TOAST classification for standardized etiology classification [29].

Data collected included demographic data (age, sex), clinical data (neurological exam at admission and during the hospitalization, NIHSS at admission, NIHSS at discharge, mRS before admission, mRS at discharge), paraclinical data at admission (complete blood count, lipid profile, INR, glycemia, fatty acids–AA, DHA, EPA).

We decided to create two groups for both stroke severity at admission as well as at discharge as follows: the group of patients with low severity (which contains the patients with NIHSS 0–4) and the group of patients with moderate–severe severity (which contains the patients with NIHSS > 5).

### 4.3. Fatty Acids Analysis

#### 4.3.1. Blood Sampling

After signing the informed consent, the blood samples were collected in the first 24 h after admission in EDTA vacutainers with a gel separator. After the blood collection, the samples were left at room temperature for 30 min for clot formation, followed by 15 min of centrifugation at 3500 rpm. Afterward, the plasma was extracted and stored in cryotubes at −40 °C until the analysis was performed.

#### 4.3.2. Analytical Method

We used a method consisting of high-performance liquid chromatography (HPLC) coupled with mass spectrometry (MS) for the quantification of AA, EPA, and DHA from the plasma samples, using the arachidonic-d11 acid as an internal standard (ST-ISTD-1).

After extracting the analytes, the samples were analyzed using reversed-phase liquid chromatography (LC) with isocratic elution and detected using specific transitions in the mass spectrometer’s MRM MS/MS module after the ionization through the negative electrospray ionization source (ESI-). The development and validation of the LC-MS/MS analytical method have not been published yet and are part of a separate, future manuscript.

#### 4.3.3. Preparation of the Standard Solutions and the Samples

##### Preparation of the Standard Solutions

The calibration standards were freshly prepared on the day of the analysis from working solutions in formic acid (0.2%). The concentration range of the standard calibration solutions was 2.5–250 µg/mL for AA and 50–2500 ng/mL for DHA and EPA.

Volumes of 200 µL of work solution, 100 µL of internal standard 1 µg/mL arachidonic-d11 acid in acetonitrile, and 500 µL of acetonitrile were mixed in an Eppendorf tube. The solution was then vortexed for 2 min at 2000 rpm, followed by 10,000 rpm centrifugation for 10 min. The supernatant was transferred into HPLC vials to be injected into the LC-MS/MS system.

##### Plasma Samples Processing

The samples are prepared fresh on the day they were unfrosted and are introduced into the machine’s autosampler immediately after processing.

We added 200 µL of patient plasma, 100 µL of internal standard, and 500 µL of acetonitrile in an Eppendorf tube. The solution was then vortexed for 2 min at 2000 rpm, followed by 10,000 rpm centrifugation for 10 min. The supernatant was transferred into HPLC vials to be injected into the LC-MS/MS system.

#### 4.3.4. LC-MS/MS Method Description

##### HPLC Method

A Perkin–Elmer Flexar 10 UHPLC system was used, and the following chromatographic conditions were applied: chromatographic column Kinetex XB-C18 3.0 × 100 mm, 2.6 µm; mobile phase 15% solution A—ammonium formate 10 mM and 85% solution B—acetonitrile; flowrate 0.400 mL/min; time of analysis 6 min; column thermostat temperature 25 °C; samples thermostat temperature 20 °C; injection volume 5 µL.

##### MS Method

The mass spectrometric detection was achieved with a Q-TOF 4600 Sciex system. After negative ion spray ionization, the analytes and internal standard were detected based on specific fragmentation patterns: the sum of ions *m*/*z* 234.94 and *m*/*z* 259.27 formed from the fragmentation of parent ion *m*/*z* 303.25 at a collision energy of −16 V for AA; the sum of ions *m*/*z* 203.19 and *m*/*z* 257.25 formed from the fragmentation of parent ion *m*/*z* 301.15 at a collision energy of −16 V for EPA; the sum of ions *m*/*z* 229.22, *m*/*z* 249.21 and *m*/*z* 283.26 formed from the fragmentation of parent ion *m*/*z* 327.25 at a collision energy of −13 V for DHA; ion *m*/*z* 270.35 formed from the fragmentation of parent ion *m*/*z* 314.26 at a collision energy of −16 V for the internal standard.

#### 4.3.5. Calibration

A linear calibration model composed of 6 levels for AA and 5 for DHA and EPA using 1/y2 weighing was applied.

### 4.4. Graphics

The graphical figures were created using the following software/online applications: www.sankeymatic.com (accessed on 7 August 2022) online application and Adobe Photoshop CS4.

### 4.5. Statistical Analysis

The data were described as continuous by the mean and standard deviation (SD) and by median and min/max, depending on the distribution. The assessment of parametric variables was performed by ANOVA test. *p*-value was set at ≤0.05 for significance. The correlation between the quantitative variables was performed using the Pearson correlation coefficient (rho), set at alpha = 0.05.

To evaluate the correlation between the distributions of the categorical variables, we used contingency tables and the Chi^2^ test.

The binary logistic regression model was based on the results obtained from the ANOVA tests, the Pearson correlation, and the crosstabs analysis of the dependent variable represented by stroke severity at discharge. The purpose of this model was to predict the patient’s functional outcome based on clinical, paraclinical, and evolutive criteria collected during the hospitalization. The model’s predictive capacity was analyzed using IBM SPSS 26 with a cut-off value of 0.5. Goodness-to-fit of the model was validated with the Hosmer–Lemeshow test.

To create the model, we used the following coded parameters with predictive potential for the outcome: age, sex (male = 1; female = 2), vascularization territory affected (carotid = 1; vertebrobasilar = 2), hemorrhagic transformation (YES = 1; NO = 2), revascularization therapy (YES = 1; NO = 2), stroke severity at admission (low severity = 1; moderate-severe severity = 2), mRS before admission, AA, DHA, EPA, DHA/EPA, DHA/AA, EPA/AA, AA/(DHA + EPA).

The dependent variable was coded as low severity with a value of 1 and moderate-severe severity with a value of 2.

The statistical analyses were performed using IBM SPSS Statistics v26 and Microsoft Excel 2019.

## 5. Conclusions

The lipid profile is and will remain an important risk factor for ischemic stroke. Even so, more and more proofs of a lipid profile paradox emerge in literature and are according to the findings in our study.

Total cholesterol and triglycerides appear to have an inverse relationship with the NIHSS at admission, with lower values associated with increased stroke severity.

Fatty acids can be used as an extension of the classic lipid profile. DHA/EPA ratio seems to be a novel biomarker that could prove useful. DHA/EPA ratio directly correlates with clinical prognostic markers such as mRS at discharge and NIHSS at admission and discharge. This biomarker was not studied before, but it appears to have great potential for future studies regarding the prognostic in ischemic stroke.

An increased EPA value in the plasma could be helpful to harvest the neuroprotective effects shown by DHA/EPA ratio and EPA/AA ratio and could prove the usefulness of EPA-enriched food supplements in future studies.

In conclusion, the impact of the lipid profile paradox and the dependency relationship with the fatty acids represent a significant predictive factor for the functional and disability prognostic of ischemic stroke patients.

## Figures and Tables

**Figure 1 ijms-23-10810-f001:**
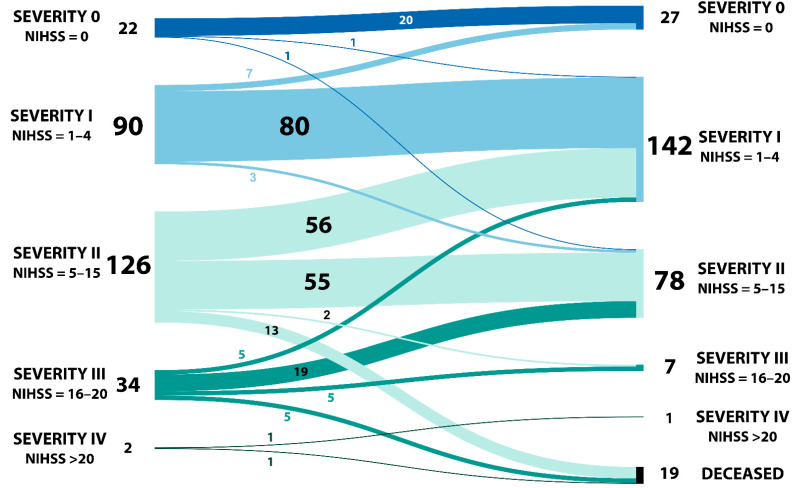
Patients’ distribution and pathway grouped by NIHSS at admission and discharge.

**Figure 2 ijms-23-10810-f002:**
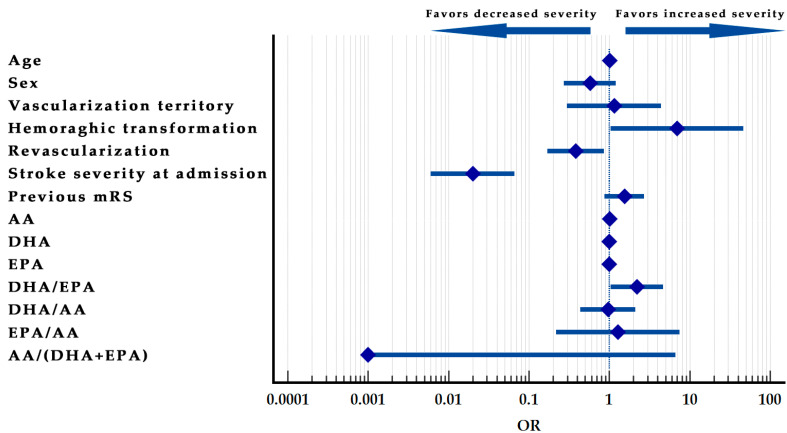
Stroke severity at discharge prediction model–Forest plot.

**Table 1 ijms-23-10810-t001:** Clinical and demographical characteristics.

Variable (n = 298)	Count	Percentage	Mean ± Standard Deviation (Min–Max)
Sex			
Male	153	51.3%	
Female	145	48.7%	
Age (years)			69.93 ± 13.45 (25–100)
Admission days			9.55 ± 6.72 (1–55)
Stroke type			
Infarct	274	91.9%	
Transient ischemic attack (TIA)	24	8.1%	
Hemorrhagic transformation			
Yes	13	4.4%	
No	285	95.6%	
Vascularization territory			
Carotid	242	81.2%	
Vertebrobasilar	56	18.8%	
Revascularization treatment			
Yes	61	20.5%	
No	237	79.5%	
TOAST classification			
Large vessel disease	49	16.4%	
Small vessel disease	28	9.4%	
Cardioembolic	89	29.9%	
Other determined etiology	12	4.0%	
Undetermined etiology	120	40.3%	

**Table 2 ijms-23-10810-t002:** Paraclinical characteristics.

Variable (n = 298)	Mean ± Standard Deviation
Fatty acids	
AA (µg/mL)	248.03 ± 117.91
DHA (ng/mL)	1168.83 ± 788.49
EPA (ng/mL)	428.40 ± 360.77
DHA/AA (‰)	4.86 ± 2.23
EPA/AA (‰)	1.77 ± 1.16
DHA/EPA	2.99 ± 0.95
AA/(DHA + EPA)	179.08 ± 76.38
Lipid profile	
Cholesterol (mg/dL)	187.40 ± 50.30
Triglycerides (mg/dL)	137.68 ± 93.61
HDL-cholesterol (mg/dL)	49.11 ± 15.17
LDL-cholesterol (mg/dL)	118.57 ± 37.52
Blood count and biochemistry	
Leucocyte count (/mm^3^)	8882.85 ± 3140.47
Hemoglobin (g/dL)	13.78 ± 1.73
Platelet count (/mm^3^)	247,359.06 ± 88,619.61
INR	1.05 ± 0.22
Glycemia (mg/dL)	129.19 ± 53.68

**Table 3 ijms-23-10810-t003:** Functional and disability patients’ data.

Variable	Count	Percentage	Mean ± Standard Deviation (Min–Max)
Rankin before admission (n = 298)			0.24 ± 0.68 (0–4)
0	258	86.6%	
1	16	5.4%	
2	18	6.0%	
3	4	1.3%	
4	2	0.7%	
5	0	0%	
Rankin at discharge (n = 298)			2.50 ± 1.85 (0–6)
0	57	19.1%	
1	44	14.8%	
2	60	20.1%	
3	40	13.4%	
4	44	14.8%	
5	34	11.4%	
6	19	6.4%	
NIHSS at admission (n = 298)			6.75 ± 5.76 (0–22)
NIHSS at discharge (n = 279)			4.28 ± 4.55 (0–21)
Infarct stroke severity at admission (n = 274)			
Low	112	40.9%	
Moderate-severe	162	59.1%	
Infarct stroke severity at discharge (n = 255)			
Low	169	66.3%	
Moderate-severe	86	33.7%	
Deceased during admission (n = 298)			
Yes	19	6.4%	
No	279	93.6%	

**Table 4 ijms-23-10810-t004:** ANOVA general data.

	Sex		*p*-Value
	Male	Female	
Age (years)	67.27 ± 14.14	72.74 ± 12.11	<0.001
Hospitalization days	8.40 ± 3.98	10.77 ± 8.57	0.002
DHA/EPA	2.84 ± 0.88	3.15 ± 1.00	0.005
Total cholesterol (mg/dL)	180.32 ± 42.82	194.87 ± 56.35	0.013
HDL-cholesterol (mg/dL)	45.61 ± 11.86	53.20 ± 17.60	0.027
Hemoglobin (g/dL)	14.42 ± 1.49	13.10 ± 1.72	<0.001
	Ischemic stroke type		
	Infarct	TIA	
Age (years)	70.53 ± 12.63	63.08 ± 19.77	0.009
Hospitalization days	9.92 ± 6.86	5.42 ± 1.99	0.002
DHA/EPA	3.04 ± 0.95	2.39 ± 0.71	0.001
	Vascularization territory		
	Carotid	Vertebro-basilar	
Age (years)	71.30 ± 13.28	64.04 ± 12.69	<0.001
Hospitalization days	10.00 ± 7.27	7.63 ± 2.77	0.017
Hemoglobin (g/dL)	13.68 ± 1.76	14.20 ± 1.52	0.042
	Hemorrhagic transformation		
	YES	NO	
Hospitalization days	20.08 ± 11.96	9.07 ± 5.99	<0.001
Leucocyte count (/mm^3^)	117,03.08 ± 5625.57	8754.21 ± 2931.92	0.001
	Large vessel disease		
	YES	NO	
Glycemia (mg/dL)	145.03 ± 90.18	126.07 ± 42.63	0.024
	Cardioembolic		
	YES	NO	
Age (years)	75.99 ± 11.96	67.35 ± 13.25	<0.001
Total cholesterol (mg/dL)	174.71 ± 49.75	192.77 ± 49.76	0.005
Triglycerides (mg/dL)	111.09 ± 44.63	148.82 ± 105.85	0.002
Hemoglobin (g/dL)	13.17 ± 1.85	14.04 ± 1.62	<0.001
INR	1.14 ± 0.28	1.01 ± 0.18	<0.001
	Stroke severity at admission		
	Low	Moderate-severe	
Age (years)	67.46 ± 12.54	72.65 ± 12.29	0.001
Hospitalization days	7.35 ± 3.01	11.16 ± 8.12	<0.001
DHA/EPA	2.90 ± 0.89	3.14 ± 0.98	0.043
	Stroke severity at discharge		
	Low	Moderate-severe	
Age (years)	69.38 ± 12.72	73.05 ± 12.11	0.026
Hospitalization days	7.92 ± 4.22	14.28 ± 9.17	<0.001
DHA/EPA	2.94 ± 0.88	3.26 ± 1.07	0.012
	Deceased during hospitalization		
	YES	NO	
Age (years)	80.11 ± 9.58	69.24 ± 13.41	0.001
Leucocyte count (/mm^3^)	10,578.42 ± 3044.88	8767.38 ± 3118.73	0.015
Hemoglobin (g/dL)	12.80 ± 1.83	13.84 ± 1.71	0.011

**Table 5 ijms-23-10810-t005:** ANOVA prognosis data.

		mRS before Admission	*p*-Value	mRS at Discharge	*p*-Value	NIHSS at Admission	*p*-Value	NIHSS at Discharge	*p*-Value
Sex	Male	0.24 ± 0.61	0.995	2.30 ± 1.73	**0.061**	6.20 ± 5.24	0.091	3.74 ± 3.99	**0.042**
	Female	0.24 ± 0.74		2.70 ± 1.70		7.33 ± 6.24		4.85 ± 5.04	
Vascularization territory	Carotid	0.25 ± 0.66	0.740	2.71 ± 1.80	**<0.001**	7.63 ± 5.67	**<0.001**	4.87 ± 4.62	**<0.001**
	Vertebrobasilar	0.21 ± 0.75		1.55 ± 1.80		2.95 ± 4.51		1.75 ± 3.24	
Hemorrhagic transformation	YES	0.38 ± 0.76	0.441	4.15 ± 1.40	**0.001**	11.62 ± 4.63	**0.002**	8.64 ± 4.36	**0.001**
	NO	0.24 ± 0.68		2.42 ± 1.84		6.53 ± 5.72		4.10 ± 4.48	
Large vessel disease	YES	0.29 ± 0.79	0.622	2.94 ± 1.72	0.068	7.80 ± 5.54	0.166	5.18 ± 4.36	0.149
	NO	0.23 ± 0.66		2.41 ± 1.87		6.55 ± 5.79		4.11 ± 4.58	
Small vessel disease	YES	0.11 ± 0.31	0.275	1.68 ± 1.46	**0.014**	3.93 ± 4.03	**0.006**	2.71 ± 3.00	0.055
	NO	0.26 ± 0.71		2.58 ± 1.87		7.04 ± 5.84		4.45 ± 4.66	
Cardioembolic	YES	0.25 ± 0.71	0.927	2.93 ± 1.99	**0.008**	8.93 ± 6.42	**<0.001**	5.21 ± 5.20	**0.030**
	NO	0.24 ± 0.67		2.31 ± 1.76		5.82 ± 5.20		3.90 ± 4.22	

**Table 6 ijms-23-10810-t006:** Crosstabs between stroke severity and clinical parameters.

		Stroke Severity at Admission			Stroke Severity at Discharge		
		Low	Moderate-Severe	*p*-Value	Low	Moderate-Severe	*p*-Value
Sex	Male	59	84	0.900	106	37	**0.050**
	Female	53	78		82	49	
Hemorrhagic transformation	YES	1	12	**0.017**	4	9	**0.005**
	NO	111	150		184	77	
Vascularization territory	Carotid	74	150	**<0.001**	144	80	**0.001**
	Vertebrobasilar	38	12		44	6	
Large vessel disease	YES	17	32	0.420	31	18	0.398
	NO	95	130		157	68	
Small vessel disease	YES	16	9	**0.018**	21	4	0.112
	NO	96	153		167	82	
Cardioembolic	YES	24	60	**0.008**	55	29	0.482

**Table 7 ijms-23-10810-t007:** Pearson correlations of all parameters and prognostic parameters.

	mRS before Admission		mRS at Discharge		NIHSS at Admission		NIHSS at Discharge	
	r	*p*-Value	r	*p*-Value	r	*p*-Value	r	*p*-Value
Age	0.058	0.317	0.386	**<0.001**	0.236	**<0.001**	0.230	**<0.001**
Hospitalization days	0.016	0.780	0.442	**<0.001**	0.458	**<0.001**	0.604	**<0.001**
AA	0.002	0.977	0.038	0.511	0.011	0.849	0.098	0.102
DHA	0.016	0.777	0.020	0.737	−0.051	0.376	−0.004	0.942
EPA	−0.025	0.664	−0.051	0.377	−0.096	0.097	−0.064	0.288
DHA/AA	−0.014	0.814	−0.029	0.617	−0.052	0.375	−0.079	0.186
EPA/AA	−0.067	0.247	−0.134	**0.020**	−0.131	**0.023**	−0.161	**0.007**
DHA/EPA	0.074	0.203	0.224	**<0.001**	0.215	**<0.001**	0.225	**<0.001**
AA/(DHA + EPA)	−0.016	0.787	−0.026	0.659	0.063	0.280	0.029	0.628
Leucocyte count	−0.009	0.879	0.219	**<0.001**	0.168	**0.004**	0.152	**0.011**
Hemoglobin	−0.070	0.227	−0.206	**<0.001**	−0.172	**0.003**	−0.106	0.078
Platelet count	−0.030	0.606	−0.036	0.534	0.009	0.878	0.025	0.681
INR	0.141	0.015	−0.029	0.620	0.018	0.761	0.003	0.955
Glycemia	0.002	0.970	0.014	0.806	−0.032	0.585	−0.104	0.083
Total cholesterol	−0.079	0.177	−0.053	0.360	−0.120	**0.040**	−0.022	0.718
Triglycerides	−0.075	0.212	−0.066	0.269	−0.122	**0.041**	−0.026	0.670
HDL-cholesterol	−0.122	0.287	−0.118	0.304	−0.120	0.294	−0.075	0.539
LDL-cholesterol	−0.013	0.910	−0.036	0.755	−0.066	0.567	−0.039	0.750
mRS before admission	1	---	0.282	**<0.001**	0.216	**<0.001**	0.203	0.001
mRS at discharge	0.282	**<0.001**	1	---	0.683	**<0.001**	0.817	**<0.001**
NIHSS at admission	0.216	**<0.001**	0.683	**<0.001**	1	**<0.001**	0.749	**<0.001**
NIHSS at discharge	0.203	**<0.001**	0.817	**<0.001**	0.749	**<0.001**	1	---

**Table 8 ijms-23-10810-t008:** Pearson correlations between paraclinical and clinical parameters.

	Age		Hospitalization Days	
	r	*p*-Value	r	*p*-Value
Age	1	---	0.222	**<0.001**
Hospitalization days	0.222	**<0.001**	1	---
AA	0.013	0.830	0.009	0.877
DHA	0.013	0.823	0.008	0.895
EPA	−0.062	0.285	0.002	0.907
DHA/AA	−0.003	0.953	0.026	0.659
EPA/AA	−0.101	0.081	−0.007	0.910
DHA/EPA	0.236	**<0.001**	0.116	**0.045**
AA/(DHA + EPA)	−0.065	0.264	−0.038	0.514
Leucocyte count	0.015	0.797	0.141	**0.015**
Hemoglobin	−0.261	**<0.001**	−0.102	0.079
Platelet count	−0.131	0.023	0.053	0.360
INR	0.054	0.356	0.001	0.991
Glycemia	0.042	0.470	−0.047	0.415
Total cholesterol	−0.051	0.385	−0.067	0.247
Triglycerides	−0.158	**0.008**	−0.056	0.353
HDL-cholesterol	−0.152	0.185	−0.037	0.748
LDL-cholesterol	−0.036	0.757	0.014	0.903

**Table 9 ijms-23-10810-t009:** Pearson correlations between fatty acids and all paraclinical parameters.

	AA		DHA		EPA		DHA/AA		EPA/AA		DHA/EPA		AA/(DHA + EPA)	
	r	*p*-Value	r	*p*-Value	r	*p*-Value	r	*p*-Value	r	*p*-Value	r	*p*-Value	r	*p*-Value
**AA**	**1**	**---**	0.610	**<0.001**	0.520	**<0.001**	**−0.142**	0.014	−0.083	0.154	−0.064	0.272	0.175	0.002
**DHA**	0.610	**<0.001**	1	---	**0.803**	**<0.001**	0.602	**<0.001**	0.450	**<0.001**	0.105	0.070	−0.458	**<0.001**
**EPA**	0.520	**<0.001**	0.803	**<0.001**	1	---	0.501	**<0.001**	0.744	**<0.001**	−0.332	**<0.001**	−0.382	**<0.001**
**DHA/AA**	−0.142	**0.014**	0.602	**<0.001**	0.501	**<0.001**	1	---	0.738	**<0.001**	0.177	**0.002**	−0.800	**<0.001**
**EPA/AA**	−0.083	0.154	0.450	**<0.001**	0.744	**<0.001**	0.738	**<0.001**	1	---	−0.403	**<0.001**	−0.590	**<0.001**
**DHA/EPA**	−0.064	0.272	0.105	0.070	−0.332	**<0.001**	0.177	**0.002**	−0.403	**<0.001**	1	---	−0.133	**0.022**
**AA/(DHA + EPA)**	0.175	**0.002**	−0.458	**<0.001**	−0.382	**<0.001**	−0.800	**<0.001**	−0.590	**<0.001**	−0.133	**0.022**	1	---
**Leucocytes**	0.048	0.414	0.067	0.247	0.100	0.086	0.020	0.734	0.050	0.387	0.003	0.966	−0.002	0.979
**Hemoglobin**	0.049	0.395	0.027	0.638	0.045	0.444	0.006	0.914	0.059	0.313	−0.139	**0.017**	0.003	0.959
**Platelets**	0.021	0.717	−0.059	0.314	−0.039	0.498	−0.085	0.141	−0.069	0.237	0.010	0.865	0.129	**0.026**
**INR**	0.041	0.487	−0.031	0.591	−0.009	0.883	−0.055	0.348	−0.027	0.649	−0.045	0.440	−0.008	0.888
**Glycemia**	0.034	0.559	0.031	0.589	0.056	0.335	0.000	0.998	0.027	0.648	0.014	0.807	−0.012	0.837
**Total cholesterol**	0.057	0.327	0.080	0.168	0.085	0.143	0.052	0.369	0.101	0.084	−0.115	**0.047**	−0.072	0.219
**Triglycerides**	0.067	0.263	0.096	0.108	0.057	0.341	0.085	0.155	0.055	0.362	−0.013	0.826	−0.069	0.250
**HDL-cholesterol**	0.075	0.514	0.210	0.065	0.203	0.075	0.309	**0.006**	0.155	0.174	−0.044	0.701	−0.170	0.138
**LDL-cholesterol**	−0.035	0.759	0.013	0.909	−0.051	0.657	0.075	0.518	−0.034	0.766	0.066	0.568	−0.083	0.471

**Table 10 ijms-23-10810-t010:** Pearson correlation between other paraclinical parameters.

		Leucocytes	Hemoglobin	Platelets	INR	Glycemia	Cholesterol	Triglycerides	HDL-c	LDL-c
Leucocytes	r	1	0.144	0.283	0.013	0.072	0.018	0.013	−0.107	0.000
	*p*-value	---	**0.013**	**<0.001**	0.822	0.216	0.764	0.833	0.350	0.997
Hemoglobin	r	0.144	1	−0.019	−0.140	0.032	0.156	0.064	0.108	0.213
	*p*-value	**0.013**	---	0.744	**0.016**	0.583	**0.007**	0.288	0.348	0.062
Platelets	r	0.283	−0.019	1	0.021	−0.033	−0.001	−0.052	0.164	−0.096
	*p*-value	**<0.001**	0.744	---	0.715	0.566	0.981	0.382	0.151	0.405
INR	r	0.013	−0.140	0.021	1	−0.001	−0.164	−0.073	−0.092	−0.132
	*p*-value	0.822	**0.016**	0.715	---	0.989	**0.005**	0.224	0.426	0.255
Glycemia	r	0.072	0.032	−0.033	−0.001	1	0.126	0.185	0.001	0.090
	*p*-value	0.216	0.583	0.566	0.989	---	**0.030**	**0.002**	0.993	0.439
Totalcholesterol	r	0.018	0.156	−0.001	−0.164	0.126	1	0.326	0.149	0.868
	*p*-value	0.764	**0.007**	**0.981**	**0.005**	**0.030**	**---**	**<0.001**	**0.195**	**<0.001**
Triglycerides	r	0.013	0.064	−0.052	−0.073	0.185	0.326	1	−0.347	0.289
	*p*-value	0.833	0.288	0.382	0.224	**0.002**	**<0.001**	---	**0.003**	**0.013**
HDLcholesterol	r	−0.107	0.108	0.164	−0.092	0.001	0.149	−0.347	1	0.127
	*p*-value	0.350	0.348	0.151	0.426	0.993	0.195	**0.003**	---	0.274
LDLcholesterol	r	0.000	0.213	−0.096	−0.132	0.090	0.868	0.289	0.127	1
	*p*-value	0.997	0.062	0.405	0.255	0.439	**<0.001**	**0.013**	0.274	---

**Table 11 ijms-23-10810-t011:** Stroke severity at discharge prediction model.

	B	SE.	Wald	df	Sig.	Exp(B)	Lower	Upper
Age	0.004	0.016	0.078	1	0.780	1.004	0.973	1.037
Sex	−0.555	0.378	2.153	1	0.142	0.574	0.274	1.205
Vascularization territory	0.137	0.689	0.040	1	0.842	1.147	0.298	4.423
Hemorrhagic transformation	1.939	0.970	4.000	1	0.046	6.954	1.040	46.524
Revascularization	−0.960	0.416	5.325	1	**0.021**	0.383	0.169	0.865
Stroke severity at admission	−3.931	0.622	39.924	1	**0.000**	0.020	0.006	0.066
mRS before admission	0.432	0.289	2.225	1	0.136	1.540	0.873	2.715
AA	0.009	0.004	5.590	1	**0.018**	1.009	1.002	1.017
DHA	−0.002	0.001	3.239	1	0.072	0.998	0.995	1.000
EPA	0.002	0.003	0.346	1	0.557	1.002	0.996	1.007
DHA/EPA	0.792	0.384	4.247	1	**0.039**	2.207	1.040	4.685
DHA/AA	−0.040	0.403	0.010	1	0.921	0.961	0.436	2.117
EPA/AA	0.243	0.901	0.073	1	0.787	1.275	0.218	7.454
AA/(DHA + EPA)	−9.127	5.620	2.637	1	0.104	0.000	0.000	6.612

## Data Availability

The data presented in this study are available on request from the corresponding author. The data are not publicly available due to institutional privacy restrictions.

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
