# Peer review of "Fatty Acids and Lipid Paradox-Neuroprotective Biomarkers in Ischemic Stroke"

_ijms, 2022, doi:10.3390/ijms231810810_

Round 1

Reviewer 1 Report

The study by Dr. S. Andone and colleagues is aimed at exploring the predictive role of certain fatty acids levels in the stroke outcome keeping in mind their possible neuroprotective effects. They prospectively analyzed over 300 consecutive patients with suspected ischemic stroke/TIA using their clinical and biochemical variables. Of the latter, docosahexaenoic acid (DHA)/ eicosapentaenoic acid (EPA) ratio, and EPA/ arachidonic acid (AA) ratio demonstrated significant positive and negative correlation, respectively, with the functional and clinical outcome variables, that may implicate “potential neuroprotective role for EPA”. The results seem to be relevant to the data and methods, however, there are several issues.

It is stated in the Introduction section, that the aim of the study is the investigation of serum ratio of certain lipid profile variables “as markers to determine the incidence of major cardiovascular events and the neurological worsening after an acute ischemic stroke”, however, this study was not designed to access incidence of any cardiovascular events, and the was no definition of what is counted as “neurological worsening” to make any implications as a result of data analysis.

Selection of patients need to be explained in more expanded way, including, i.e., time from stroke onset to admission and blood sampling (to ensure that data was collected for an acute stroke phase), previous statin treatment, etc.

Precise definitions of outcomes and variables are crucial, and this study lacks it. The model that is used in the study includes “age, sex, hospitalization days, vascularization territory affected, hemorrhagic transformation, and DHA/EPA ratio”. However, there are several established predictors of stroke outcome (e.g., https://www.ahajournals.org/doi/10.1161/STROKEAHA.120.033785), with the baseline stroke severity and revascularization being ones that may have the strongest impact. This might be subject for statistical consultation, as the study focuses only on certain metabolic variables and it is not aimed at and powered to assess other possible predictors.

The description of the results includes a very detailed report of the data, that might not be relevant to the main point of the study.

In summary, general recommendations include statistic consultation, more clear presentation of the baseline patient characteristics and results focusing on the data relevant to the research question and improvement of the Discussion section by presenting the summary of the results in the context of related studies, indicating strengths and limitations of the research, that is currently missing. Also, proof reading might be useful (e.g., “carotidian” throughout the text, “imagistic criteria” in line 405, etc.). EPA/AA, DHA/EPA abbreviations in the abstract need to be explained.

Author Response

The study by Dr. S. Andone and colleagues is aimed at exploring the predictive role of certain fatty acids levels in the stroke outcome keeping in mind their possible neuroprotective effects. They prospectively analyzed over 300 consecutive patients with suspected ischemic stroke/TIA using their clinical and biochemical variables. Of the latter, docosahexaenoic acid (DHA)/ eicosapentaenoic acid (EPA) ratio, and EPA/ arachidonic acid (AA) ratio demonstrated significant positive and negative correlation, respectively, with the functional and clinical outcome variables, that may implicate “potential neuroprotective role for EPA”. The results seem to be relevant to the data and methods, however, there are several issues.

  1. It is stated in the Introduction section, that the aim of the study is the investigation of serum ratio of certain lipid profile variables “as markers to determine the incidence of major cardiovascular events and the neurological worsening after an acute ischemic stroke”, however, this study was not designed to access incidence of any cardiovascular events, and the was no definition of what is counted as “neurological worsening” to make any implications as a result of data analysis.

This was modified accordingly to the correct purpose of the study.

  1. Selection of patients need to be explained in more expanded way, including, i.e., time from stroke onset to admission and blood sampling (to ensure that data was collected for an acute stroke phase), previous statin treatment, etc.

We added additional information regarding the inclusion criteria for the patients to address the mentioned problems (time from onset, previous treatment, revascularization therapy).

As for the blood sampling, it was already mentioned in the 4.3.1 subsection that: “After signing the informed consent, the blood samples were collected in the first 24 hours after admission”.

  1. Precise definitions of outcomes and variables are crucial, and this study lacks it. The model that is used in the study includes “age, sex, hospitalization days, vascularization territory affected, hemorrhagic transformation, and DHA/EPA ratio”. However, there are several established predictors of stroke outcome (e.g., https://www.ahajournals.org/doi/10.1161/STROKEAHA.120.033785), with the baseline stroke severity and revascularization being ones that may have the strongest impact. This might be subject for statistical consultation, as the study focuses only on certain metabolic variables and it is not aimed at and powered to assess other possible predictors.

We changed the model by adding additional variables like revascularization therapy, stroke severity at admission, modified Rankin scale (mRS) before admission and the entire fatty acids profile. We modified the table, the figure, discussion and material and method accordingly to the new model.

  1. The description of the results includes a very detailed report of the data, that might not be relevant to the main point of the study.

Indeed, we introduced additional data that are not targeted towards the main purpose of the study, but  given the fact that these variables showed potential for explaining the functional, disability and mortality outcome, we would prefer to leave them as it is.

  1. In summary, general recommendations include statistic consultation, more clear presentation of the baseline patient characteristics and results focusing on the data relevant to the research question and improvement of the Discussionsection by presenting the summary of the results in the context of related studies, indicating strengths and limitations of the research, that is currently missing.

We followed the suggestions and did the necessary adjustments.

  1. Also, proof reading might be useful (e.g., “carotidian” throughout the text, “imagistic criteria” in line 405, etc.). EPA/AA, DHA/EPA abbreviations in the abstract need to be explained.

Proof reading was performed, the mentioned problems were resolved.

Reviewer 2 Report

The study aimed to analyze the impact of the lipid profile and the serum fatty acids 265 panel (including AA, DHA, EPA, and their ratios) over the functional, disability, and mor- 266 tality outcomes in ischemic stroke patients. It is a well designed study with very high sample size, leading to a significant conclusion. The study should be published as is. The reviewer would suggest a minor question:

Abstract:

I know you have defined them in the introduction but one usually reads the abstract first. Thus, I suggest you either define or reword as follows:

- DHA, EPA: you define in the abstract or spell out. Otherwise, you describe your stat result saying that "the measures on omega-3 showed an inverse relationship ....etc.".

- NIHSS: please spell out or define. We don't know the abbreviation.

Thanks. 

Author Response

The study aimed to analyze the impact of the lipid profile and the serum fatty acids 265 panel (including AA, DHA, EPA, and their ratios) over the functional, disability, and mor- 266 tality outcomes in ischemic stroke patients. It is a well designed study with very high sample size, leading to a significant conclusion. The study should be published as is. The reviewer would suggest a minor question:

Abstract:

I know you have defined them in the introduction but one usually reads the abstract first. Thus, I suggest you either define or reword as follows:

  1. DHA, EPA: you define in the abstract or spell out. Otherwise, you describe your stat result saying that "the measures on omega-3 showed an inverse relationship ....etc.".

This was corrected.

  1. NIHSS: please spell out or define. We don't know the abbreviation.

This was corrected.

Thank you for your suggestions.

Round 2

Reviewer 1 Report

The authors clarified most of the issues.